# Nanoparticle Strategies for Treating CNS Disorders: A Comprehensive Review of Drug Delivery and Theranostic Applications

**DOI:** 10.3390/ijms252413302

**Published:** 2024-12-11

**Authors:** Corneliu Toader, Adrian Vasile Dumitru, Lucian Eva, Matei Serban, Razvan-Adrian Covache-Busuioc, Alexandru Vlad Ciurea

**Affiliations:** 1Department of Neurosurgery, “Carol Davila” University of Medicine and Pharmacy, 050474 Bucharest, Romania; corneliu.toader@umfcd.ro (C.T.); matei.serban2021@stud.umfcd.ro (M.S.); razvan-adrian.covache-busuioc0720@stud.umfcd.ro (R.-A.C.-B.); prof.avciurea@gmail.com (A.V.C.); 2Department of Vascular Neurosurgery, National Institute of Neurology and Neurovascular Diseases, 077160 Bucharest, Romania; 3Department of Pathology, “Carol Davila” University of Medicine and Pharmacy, 050474 Bucharest, Romania; 4Department of Pathology, University Emergency Hospital of Bucharest, 050098 Bucharest, Romania; 5Department of Neurosurgery, Dunarea de Jos University, 800010 Galati, Romania; 6Department of Neurosurgery, Clinical Emergency Hospital “Prof. Dr. Nicolae Oblu”, 700309 Iasi, Romania; 7Neurosurgery Department, Sanador Clinical Hospital, 010991 Bucharest, Romania; 8Medical Section Within the Romanian Academy, 010071 Bucharest, Romania

**Keywords:** blood–brain barrier, nanoparticles, neurodegenerative diseases, neuroregeneration, theranostics, Alzheimer’s disease, Parkinson’s disease, nanotechnology, targeted drug delivery, nanomedicine

## Abstract

This review aims to address the significant challenges of treating central nervous system (CNS) disorders such as neurodegenerative diseases, strokes, spinal cord injuries, and brain tumors. These disorders are difficult to manage due to the complexity of disease mechanisms and the protective blood–brain barrier (BBB), which restricts drug delivery. Recent advancements in nanoparticle (NP) technologies offer promising solutions, with potential applications in drug delivery, neuroprotection, and neuroregeneration. By examining current research, we explore how NPs can cross the BBB, deliver medications directly to targeted CNS regions, and enhance both diagnostics and treatment. Key NP strategies, such as passive targeting, receptor-mediated transport, and stimuli-responsive systems, demonstrate encouraging results. Studies show that NPs may improve drug delivery, minimize side effects, and increase therapeutic effectiveness in models of Alzheimer’s, Parkinson’s, stroke, and glioblastoma. NP technologies thus represent a promising approach for CNS disorder management, combining drug delivery and diagnostic capabilities to enable more precise and effective treatments that could significantly benefit patient outcomes.

## 1. Introduction

### 1.1. Overview of Neuroscience Challenges and Role of Nanotechnology

#### 1.1.1. Challenges in Neuroscience

Central nervous system (CNS) disorders encompass a broad range of debilitating conditions, including neurodegenerative diseases like Alzheimer’s and Parkinson’s, cerebrovascular events such as strokes, traumatic spinal cord injuries, and aggressive malignancies like glioblastoma multiforme [1]. These conditions share a common complexity, with their impact extending across millions globally and their management hindered by the intrinsic challenges of the CNS. Alzheimer’s disease, the leading cause of dementia, is projected to affect nearly twice its current global population by mid-century. Parkinson’s, driven by the progressive loss of dopaminergic neurons, is emerging as the fastest-growing neurodegenerative disease, reflecting a substantial and increasing burden [2].

Acute injuries like stroke and spinal cord trauma exemplify the intricate pathophysiology of CNS disorders. Stroke is marked by a cascade of ischemia-induced damage and secondary inflammation that leads to irreversible neuronal loss within hours. Spinal cord injuries, in contrast, create an environment hostile to regeneration through the formation of inhibitory scar tissue and persistent neuroinflammation [3]. Glioblastoma multiforme, characterized by its aggressive infiltration and resistance to apoptosis, defies conventional treatments, leaving patients with dismal survival outcomes. These disorders demand advanced solutions capable of addressing their unique and multifaceted pathologies [4].

#### 1.1.2. Limitations of Current Approaches

Despite advancements, existing diagnostic and therapeutic approaches fall short in addressing the specific demands of CNS diseases. The blood–brain barrier (BBB) acts as a near-impenetrable shield, restricting over 98% of small molecules and almost all macromolecular therapies [5,6]. This selective permeability limits the efficacy of systemic treatments, requiring higher doses that frequently result in off-target effects and toxicity. For instance, conventional chemotherapy for glioblastoma suffers from poor CNS penetration, leading to limited efficacy and significant systemic side effects [7].

Diagnostics also face critical shortcomings. Current imaging techniques, such as MRI and PET, often lack the sensitivity required for detecting early pathological changes. This delay in diagnosis results in missed opportunities for early intervention, particularly in progressive conditions like Alzheimer’s and Parkinson’s. Invasive diagnostic tools, while precise, are impractical for regular use, and non-invasive methods often fail to provide sufficient specificity to differentiate overlapping pathologies. These diagnostic and therapeutic gaps highlight the need for innovative technologies capable of precise targeting, real-time monitoring, and reducing systemic burden [8].

### 1.2. Emergence of Nanotechnology

Nanotechnology is redefining the landscape of CNS research, offering transformative potential to overcome many of these barriers. By engineering nanomaterials with tunable properties at the scale of 1 to 100 nanometers, researchers have unlocked the ability to precisely target disease sites, enhance therapeutic delivery, and improve diagnostic sensitivity. Unlike conventional systems, nanoparticles (NPs) can be designed for targeted transport across the BBB, functionalized with ligands for cell-specific delivery, and triggered to release therapeutic payloads in response to local stimuli, such as enzymatic activity or pH changes [9,10].

In therapeutic applications, NPs have shown remarkable versatility. Functionalized NPs can exploit receptor-mediated transport mechanisms at the BBB, enabling active targeting and minimizing off-target toxicity. For example, receptor-specific ligands can direct NPs carrying chemotherapeutics to glioblastoma cells while sparing healthy brain tissue. Similarly, stimuli-responsive NPs have demonstrated precise drug release in the microenvironment of CNS tumors, reducing systemic exposure and enhancing efficacy [11].

Diagnostics are also benefiting significantly from nanotechnology. NPs functionalized with imaging agents enhance the resolution and specificity of modalities like MRI and PET. For example, magnetic NPs improve contrast in MRI scans, enabling the detection of minute structural changes associated with neurodegenerative diseases [12]. Quantum dots and other nanoscale fluorescent agents allow real-time tracking of biological processes, providing insights into disease progression at a molecular level. Emerging nanoparticle-based biosensors are enabling non-invasive detection of CNS biomarkers in peripheral fluids, significantly improving accessibility and reducing dependence on invasive sampling techniques [13].

One of the most promising innovations in nanotechnology is theranostics, where therapeutic and diagnostic functions are integrated into a single platform. Theranostic NPs enable simultaneous detection, drug delivery, and monitoring of therapeutic efficacy. Recent studies have demonstrated their potential in preclinical CNS models, combining precision imaging with site-specific treatment to address diseases like glioblastoma and stroke [14]. This convergence of diagnostic and therapeutic capabilities underscores the transformative role of nanotechnology in neuroscience [15].

This review focuses on the latest advancements in nanoparticle-based approaches for CNS applications, with an emphasis on overcoming the challenges of drug delivery, enhancing diagnostic capabilities, and enabling regenerative therapies. It will explore the potential of nanotechnology to reshape the diagnosis, treatment, and management of CNS disorders, while addressing the critical challenges and future directions for clinical translation. By focusing on novel strategies and their scientific impact, this review aims to highlight nanotechnology as a cornerstone of future neuroscience innovations.

## 2. NPs for Drug Delivery Across the Blood–Brain Barrier

### 2.1. Mechanisms of BBB Penetration

#### Challenges Posed by the BBB

The BBB is a sophisticated protective structure that selectively regulates the exchange of substances between the circulatory system and the CNS. Comprised of endothelial cells with tight junctions, pericytes, astrocytic end-feet, and a basal lamina, the BBB restricts the passage of nearly all therapeutic macromolecules and over 98% of small-molecule drugs [16]. Furthermore, active efflux transporters like P-glycoprotein (P-gp) and multidrug resistance proteins expel compounds that penetrate endothelial cells, limiting drug retention within the brain [17]. This creates a formidable challenge for treating CNS diseases such as glioblastoma, Alzheimer’s, and Parkinson’s, where localized and consistent therapeutic delivery is critical [5].

Despite these challenges, disease-specific BBB disruptions offer unique opportunities for targeted delivery. For example, the blood–tumor barrier (BTB) in glioblastoma exhibits regions of enhanced permeability, enabling selective nanoparticle accumulation. Similarly, localized dysfunctions in neurodegenerative diseases allow NPs to access affected regions more effectively. Leveraging these pathological alterations, nanotechnology has emerged as a transformative strategy for addressing the limitations of conventional CNS therapies [18].

### 2.2. Nanoparticle Strategies for BBB Penetration

#### 2.2.1. Passive Targeting

Passive targeting exploits the physicochemical properties of NPs to achieve BBB penetration without requiring specific interactions with receptors or transporters. NPs under 100 nm with optimized surface properties, such as lipophilic coatings or PEGylation, can diffuse across the BBB via adsorptive-mediated transcytosis or nonspecific interactions [19]. Lipid NPs, including liposomes and solid lipid NPs (SLNs), excel in this domain due to their stability and ability to encapsulate both hydrophilic and hydrophobic drugs. Recent advances in these systems include modifications for extended circulation and targeted release in diseased CNS regions [20]. Figure 1 highlights the blood–brain barrier’s (BBB) intricate interplay with endothelial cells, neurons, and infiltrating immune cells. By depicting how alterations in BBB integrity can facilitate immune cell migration, it underscores the potential progression from localized inflammation to widespread neurodegenerative changes within the central nervous system.

#### 2.2.2. Active Targeting

Active targeting utilizes receptor-mediated transcytosis (RMT) pathways to selectively transport therapeutic agents across the BBB. NPs functionalized with ligands for transferrin, insulin, or low-density lipoprotein receptors demonstrate enhanced delivery specificity [21]. Recent innovations include bispecific NPs, which simultaneously target multiple receptors, such as transferrin and integrin [22]. This dual-targeting approach has significantly improved therapeutic delivery in ischemic stroke and neurodegenerative disease models. Aptamer-functionalized NPs offer an emerging alternative, combining high specificity with low immunogenicity for advanced applications in chronic CNS conditions [23].

#### 2.2.3. Stimuli-Responsive Systems

Stimuli-responsive NPs release their therapeutic payload in response to specific environmental or external triggers, offering unmatched precision. pH-sensitive NPs, for instance, exploit the acidic microenvironment of glioblastomas to deliver chemotherapeutics directly to tumor tissues [24]. Magnetic NPs can be guided via external magnetic fields to targeted regions of the brain, where controlled hyperthermia triggers localized drug release. Additionally, ultrasound-responsive NPs combined with focused ultrasound technology enable non-invasive, localized BBB opening, expanding the therapeutic scope for both neurodegenerative and oncological conditions [25,26].

### 2.3. Case Studies

Lipid NPs are well established in CNS drug delivery, particularly for their biocompatibility and ability to encapsulate diverse therapeutic agents. PEGylated liposomes functionalized with ApoE-mimetic peptides have achieved remarkable results in Alzheimer’s models, crossing the BBB and delivering anti-amyloid agents with high specificity [19]. SLNs, known for their stability and sustained release capabilities, have been employed in traumatic brain injury models to deliver neuroprotective agents, reducing oxidative stress and inflammation [27]. Advances in co-delivery systems using SLNs have demonstrated synergistic effects by combining chemotherapeutics with anti-inflammatory drugs, offering novel strategies for multifactorial CNS diseases [28,29]. Figure 2 provides an overview of how engineered liposomes can be guided to cancer cells through receptor-specific interactions.

Polymeric NPs (PNPs) stand out for their tunability and precision. Biodegradable polymers like poly(lactic-co-glycolic acid) (PLGA) are widely used for sustained-release formulations. Recent designs include dual-layered PNPs, delivering neuroprotective agents in outer layers and gene-editing tools in core regions, enabling sequential therapy in complex pathologies such as spinal cord injuries [30]. Chitosan-based nanogels, functionalized with receptor-targeting ligands, have shown exceptional potential in Alzheimer’s and Huntington’s models, delivering monoclonal antibodies and CRISPR-Cas9 systems with unprecedented accuracy [30].

Exosomes, naturally occurring vesicles, are gaining momentum as endogenous carriers for CNS therapeutics. Engineered exosomes loaded with siRNA have successfully silenced pathological genes in Alzheimer’s and Parkinson’s models, reducing disease-specific protein aggregates [31]. Hybrid exosome systems, integrating magnetic NPs, have been employed for stroke therapy, enabling precise localization of neuroprotective agents. These hybrid designs bridge biological functionality with advanced targeting, representing a novel frontier in nanomedicine [32].

This table (Table 1) summarizes the diverse nanoparticle types utilized for crossing the BBB, highlighting their mechanisms, therapeutic and diagnostic applications, recent innovations, and key references.

### 2.4. Emerging Trends: Hybrid NPs

Hybrid NPs combine the benefits of multiple nanocarrier systems. Lipid–polymer hybrids (LPHNs) integrate the stability of polymers with the biocompatibility of lipids, enhancing tumor targeting in glioblastoma models [47]. Metallic–polymeric hybrids, embedding magnetic or gold NPs in polymer matrices, enable simultaneous imaging and therapy. These systems have demonstrated superior performance in both diagnostics and treatment, offering new opportunities for theranostic applications [48].

The advancements in drug delivery systems across the BBB underscore the versatility and potential of NPs in overcoming the challenges of CNS therapies. However, the integration of diagnostic and therapeutic functionalities within a single nanoparticle platform—known as theranostics—marks the next frontier in nanotechnology [49]. Theranostic NPs have the potential to simultaneously deliver targeted therapies and monitor disease progression in real time, offering a dynamic approach to precision medicine. The following section will explore the dual functionality of theranostic NPs, emphasizing their role in enhancing diagnostics and treatment outcomes for neurological diseases [50].

## 3. Theranostic NPs in Neurology

### 3.1. Combining Diagnostics and Therapeutics: Applications in Neurodegenerative Diseases

#### Dual-Function NPs

TNPs represent a pivotal advancement in neurology, merging diagnostic and therapeutic modalities into a single platform. This integration addresses two critical needs: the ability to monitor disease progression in real time and to deliver targeted therapeutic interventions. Such capabilities are particularly transformative in neurodegenerative diseases like AD and PD, where pathological changes often precede clinical symptoms by years [51].

The design of TNPs focuses on leveraging nanoscale features for efficient BBB penetration and selective interaction with disease-specific targets. For instance, NPs functionalized with amyloid-beta antibodies or tau-binding peptides have demonstrated efficacy in AD models, enabling imaging of pathological aggregates alongside the delivery of anti-amyloid therapies [52]. Similarly, alpha-synuclein-targeting NPs in PD provide a dual benefit: fluorescence-guided imaging of protein aggregates and localized release of neuroprotective agents [53].

### 3.2. Applications in Neurodegeneration

Advancements in theranostic technologies for AD focus on tackling its hallmark pathologies: amyloid plaques and tau tangles. Iron oxide NPs functionalized with amyloid-specific ligands have been utilized to enhance MRI contrast, allowing early detection of plaques in preclinical models [54]. Simultaneously, these NPs deliver anti-inflammatory or anti-amyloid agents, effectively reducing plaque deposition and neuroinflammation. More recent approaches integrate PET tracers with therapeutic agents to provide multimodal insights into disease progression and treatment responses [55].

NPs delivering small interfering RNA (siRNA) targeting amyloid precursor protein (APP) have demonstrated the ability to simultaneously suppress amyloid production and provide imaging capabilities through quantum dot (QD) fluorescence [56]. These approaches are increasingly combined with oxidative stress modulators to address secondary pathologies, highlighting the multifunctionality of TNPs [57].

TNPs for PD are designed to address early dopaminergic neuronal loss and alpha-synuclein aggregation. Magnetic NPs functionalized with alpha-synuclein antibodies have been employed to visualize protein accumulation through MRI while delivering neuroprotective agents such as dopamine precursors or antioxidants [58]. Recent advances include NPs loaded with neurotrophic factors like glial cell line-derived neurotrophic factor (GDNF), promoting dopaminergic neuron survival while enabling high-resolution tracking of neuronal repair [58].

An emerging innovation involves the use of QDs conjugated with mitochondrial protective agents. These NPs facilitate the restoration of mitochondrial function, a critical factor in PD progression, while simultaneously providing real-time imaging of therapeutic efficacy [59].

This table (Table 2) explores theranostic NPs, which combine therapeutic and diagnostic functions, focusing on their unique modalities, recent advancements, and references.

### 3.3. Examples and Recent Advances in Imaging and Therapy

Iron oxide NPs have been extensively studied for their MRI-enhancing properties in CNS disorders. By conjugating these particles with BBB-penetrating peptides such as transferrin or low-density lipoprotein receptor ligands, researchers have achieved both enhanced imaging resolution and targeted delivery of neuroprotective drugs. In AD models, MRI-guided NPs loaded with curcumin have demonstrated significant reductions in amyloid plaque burden while providing longitudinal imaging of therapeutic effects [74,75].

QDs have emerged as a versatile theranostic tool due to their tunable fluorescence and exceptional photostability. In PD models, QDs functionalized with dopamine transporter ligands have been used to visualize dopaminergic neuronal activity while delivering agents that mitigate oxidative stress [76]. Advanced near-infrared (NIR) QDs now allow for deeper tissue imaging, broadening their applicability in CNS diagnostics and therapy [77].

Multimodal TNPs that combine imaging techniques like MRI, PET, and optical fluorescence provide unprecedented diagnostic accuracy. Gold-coated iron oxide NPs, for example, offer dual imaging capabilities and enhanced therapeutic delivery in glioblastoma models [78]. These NPs have been engineered to release chemotherapeutics in response to localized stimuli such as pH changes, ensuring precise treatment while tracking tumor dynamics.

### 3.4. Case Studies: Preclinical and Clinical Advances

Recent preclinical studies on TNPs in AD highlight their ability to integrate diagnostics with treatment. NPs functionalized with tau-specific antibodies and loaded with tau aggregation inhibitors have demonstrated efficacy in reducing tauopathy while enabling PET imaging of tangle dynamics [79]. These platforms are now being refined to include multi-targeting capabilities, addressing both amyloid and tau pathologies simultaneously [80].

In PD, TNPs combining optical imaging and therapeutic delivery have shown promise in restoring dopaminergic function. Gold NPs carrying dopamine biosynthesis enzymes have enabled sustained neurotransmitter production while allowing for precise tracking of drug localization through fluorescence [81]. Exosome-derived TNPs, engineered to deliver siRNA targeting alpha-synuclein, have demonstrated reductions in protein aggregation and improvements in motor function in animal models [82].

In glioblastoma, theranostic approaches are advancing toward clinical applications. Iron oxide NPs conjugated with tumor-homing peptides and loaded with chemotherapeutics have been successfully used to delineate tumor margins intraoperatively while providing sustained drug release post-surgery [83]. Multimodal platforms incorporating MRI, PET, and photoacoustic imaging are further improving the precision of tumor visualization and treatment monitoring [84].

### 3.5. Emerging Innovations in Theranostics

Hybrid TNPs, which combine the properties of multiple nanocarrier systems, are gaining traction for their multifunctionality. Lipid–polymer hybrids enhance drug encapsulation and stability while incorporating imaging agents for diagnostic applications [85]. In glioblastoma models, hybrid NPs carrying immunomodulators and chemotherapeutics have demonstrated superior tumor targeting and immune activation [86].

Biomimetic NPs, inspired by natural structures such as exosomes, offer promising advantages for theranostic applications. Engineered exosome-like NPs loaded with PET tracers and neuroprotective drugs have shown high specificity for neuronal tissues while maintaining low immunogenicity [87]. These systems are now being explored for applications in both AD and PD, addressing the dual challenges of early diagnosis and effective therapy.

The advancements in theranostic NPs highlight their transformative potential in addressing diagnostic and therapeutic challenges in CNS disorders. However, the ability of NPs to actively support neuronal survival and promote tissue repair is equally critical in conditions such as stroke and spinal cord injury, where both acute protection and long-term regeneration are essential. The following section will explore the emerging role of NPs in neuroprotection and neuroregeneration, focusing on their potential to redefine recovery strategies in CNS injuries.

## 4. Neuroprotection and Neuroregeneration

### 4.1. Potential for Regeneration After Stroke and Spinal Cord Injury

#### NPs for Neuroprotection

The aftermath of CNS injuries, such as ischemic stroke and SCI, is characterized by a cascade of secondary damage processes, including oxidative stress, inflammation, excitotoxicity, and neuronal apoptosis. These processes significantly exacerbate primary injury and hinder recovery. NPs have emerged as a promising solution to mitigate secondary damage by delivering therapeutic agents with high precision and sustained release, directly targeting affected regions [88].

Advanced antioxidant NPs, such as cerium oxide (CeO_2_) and selenium (Se) NPs, have demonstrated significant neuroprotective properties. Cerium oxide NPs, with their regenerative catalytic surface, can scavenge reactive oxygen species (ROS) persistently, mimicking the activity of natural antioxidants like superoxide dismutase and catalase [89]. In ischemic stroke models, CeO_2_ NPs have reduced infarct volumes, improved motor function, and provided long-lasting neuroprotection [90]. Selenium NPs, on the other hand, act by reducing oxidative stress while enhancing the expression of endogenous protective genes, such as nuclear factor erythroid 2-related factor 2 (Nrf2), further limiting neuronal damage [91].

Inflammation is a critical component of secondary injury, and NPs targeting inflammatory pathways are being developed to modulate the immune response. Polymeric NPs encapsulating interleukin-10 (IL-10) have shown the ability to reprogram activated microglia, reducing pro-inflammatory cytokines such as TNF-α and IL-1β while promoting an anti-inflammatory environment [92,93]. In parallel, lipid-based NPs delivering glucocorticoids such as dexamethasone have been optimized for selective microglial targeting, achieving substantial reductions in neuroinflammation with minimal systemic side effects [94].

Addressing excitotoxicity, a hallmark of acute CNS injuries, NPs have been developed to regulate excessive glutamate signaling. Lipid NPs encapsulating NMDA receptor antagonists like memantine have demonstrated targeted inhibition of excitotoxic pathways, reducing neuronal apoptosis without the off-target effects observed in systemic drug administration [95].

### 4.2. Guiding Neural Regeneration

#### Nanoparticle-Assisted Neural Repair

Regeneration following CNS injuries depends on the ability to restore structural and functional connectivity. NPs are being engineered to deliver regenerative agents such as growth factors, enzymes, and inhibitory pathway modulators with precise spatial and temporal control. For example, PLGA NPs loaded with brain-derived neurotrophic factor (BDNF) provide sustained release, enabling prolonged stimulation of axonal growth and synaptogenesis [96]. In preclinical SCI models, these NPs have facilitated significant improvements in motor recovery when combined with physical rehabilitation therapies [97].

Targeting inhibitory molecules in the extracellular matrix is another promising strategy to promote neural regeneration. Lipid-based NPs encapsulating chondroitinase ABC have been shown to degrade chondroitin sulfate proteoglycans (CSPGs), which are key inhibitors of axonal growth. These systems have enabled axonal sprouting and functional connectivity in SCI models [98]. Dual-action NPs that deliver both growth factors and CSPG-degrading enzymes have demonstrated enhanced regenerative potential by simultaneously creating a permissive environment and stimulating regrowth [99].

Bioactive NPs interact directly with the neural microenvironment to support repair and regeneration. Mesoporous silica NPs functionalized with adhesion peptides, such as RGD (Arg-Gly-Asp), have shown the ability to enhance neural stem cell migration and differentiation [100]. Calcium phosphate NPs, designed to mimic bone-like mineral structures, provide a supportive scaffold for neural progenitor cells, promoting their differentiation into functional neurons and glial cells [101].

Magnetic NPs provide an additional dimension to neural repair strategies. These particles can be guided to specific sites of injury using external magnetic fields, delivering therapeutic agents while also offering mechanical stimulation to align regenerating axons [102,103]. Studies have demonstrated that magnetically guided NPs coated with nerve growth factor (NGF) significantly enhance neurite extension and reestablish neural pathways in SCI models. Furthermore, magnetic stimulation of these particles has been shown to activate mechanosensitive ion channels, further promoting axonal regrowth and functional recovery [104].

This table (Table 3) details NPs employed in neuroprotection and neuroregeneration, focusing on their mechanisms, applications, and cutting-edge developments.

### 4.3. Role of Magnetic and Other Bioactive NPs

Magnetic NPs (MNPs) are uniquely suited for precise localization and therapeutic delivery in CNS injuries. By applying external magnetic fields, these particles can be directed to injury sites, providing targeted delivery of neuroprotective agents [119]. In addition to their targeting capabilities, MNPs coated with neurotrophic factors such as NGF or glial cell line-derived neurotrophic factor (GDNF) have demonstrated efficacy in promoting axonal alignment and enhancing neural repair [120,121].

Thermally responsive MNPs, which generate localized heat under alternating magnetic fields, are being explored for their ability to modulate cell behavior and enhance therapeutic outcomes [122]. This approach, known as magnetic hyperthermia, has been shown to increase the expression of heat shock proteins, protecting neurons from secondary injury and supporting recovery [123].

Bioactive NPs, such as hydroxyapatite and mesoporous silica systems, provide unique advantages in neural repair. Hydroxyapatite NPs, functionalized with neural adhesion molecules, promote the recruitment and differentiation of neural progenitor cells while supporting angiogenesis, a critical factor for tissue repair. Mesoporous silica NPs have been engineered to deliver dual-acting agents, supporting both neural stem cell migration and vascular regeneration at injury sites [124].

Hybrid NPs combining bioactive coatings with magnetic properties represent a new frontier in CNS repair. For instance, magnetic silica NPs functionalized with laminin peptides have been used to simultaneously support stem cell adhesion and guide axonal alignment, demonstrating synergistic effects in SCI models [125].

### 4.4. Emerging Innovations in Regenerative Nanotechnology

Exosome-mimetic NPs are a biomimetic advancement in regenerative nanotechnology. These systems replicate the structure and functionality of natural exosomes, allowing them to cross the BBB and deliver bioactive cargo with high specificity. NPs mimicking exosomal properties have been loaded with microRNAs such as miR-124, which promote neuronal differentiation and reduce glial scar formation, demonstrating substantial improvements in neural repair in SCI models [126]. Three-dimensional nanostructures integrated with NPs offer a sophisticated approach to CNS regeneration. Electrospun nanofibers combined with NPs carrying neurotrophic factors have provided structural support for axonal growth while enabling sustained therapeutic delivery. Nanogel scaffolds impregnated with regenerative NPs have further improved outcomes by promoting vascular and neural integration in SCI models [127].

The advances in nanoparticle-based neuroprotection and regeneration illustrate their transformative potential in CNS repair. However, significant barriers remain in translating these therapies into clinical practice, including challenges related to long-term safety, scalability, and regulatory approval. The following section will address these issues, exploring the path toward clinical translation while emphasizing the ethical and practical considerations required to bring these innovations to widespread use.

## 5. Current Challenges and Future Directions

### 5.1. Clinical Translation Challenges

A key challenge in translating NP-based therapies for CNS disorders into clinical use is ensuring long-term safety. While NPs have shown remarkable efficacy in preclinical studies, their interactions with neural and systemic environments remain insufficiently understood [128]. Inorganic NPs, such as iron oxide and quantum dots, are particularly scrutinized for potential accumulation in sensitive tissues, leading to oxidative stress and chronic inflammation. Even biocompatible NPs, such as those derived from lipids or polymers, can provoke unintended immune responses or disrupt homeostasis in the CNS’s tightly regulated microenvironment [129].

Emerging solutions are focused on integrating advanced biodegradability and dynamic responsiveness into nanoparticle design. Hybrid systems that combine inorganic cores with biodegradable shells allow for targeted delivery with minimal residue, reducing risks of accumulation [130]. For instance, self-degrading polymeric shells activated by environmental cues, such as pH or enzymatic activity, ensure complete clearance after therapeutic action [131]. Advances in nanotoxicology are also driving the development of predictive models to evaluate the long-term impact of NPs on neural networks and systemic health, using human-derived organoids and machine learning-based toxicity profiling [132].

The complexity of nanoparticle synthesis poses significant barriers to scaling up production while maintaining consistency. Variability in nanoparticle size, surface functionalization or encapsulation efficiency can critically affect therapeutic outcomes [133]. Traditional batch synthesis methods struggle with reproducibility at industrial scales, limiting their feasibility for widespread clinical adoption [134,135].

Continuous-flow microfluidics is emerging as a transformative technology to address these challenges, enabling high-throughput synthesis with precise control over NP properties [136]. This approach is being augmented with AI-driven optimization, which adjusts synthesis parameters in real time to ensure consistency across batches. Furthermore, modular manufacturing platforms are being explored to integrate multiple nanoparticle functionalities—such as drug delivery, imaging, and targeting—into scalable production pipelines [137].

The unique physicochemical properties of NPs necessitate the development of specialized regulatory frameworks [138]. Traditional paradigms used for small-molecule drugs or biologics fail to capture the complexities of nanoscale behaviors, such as size-dependent interactions, surface charge effects, or dynamic transformations in biological environments. This regulatory ambiguity often delays the approval process for promising nanoparticle-based therapies [139].

Regulatory bodies like the FDA and EMA are collaborating with academic and industrial stakeholders to define new standards tailored to nanomedicine. These initiatives emphasize harmonized guidelines for characterizing NP safety, efficacy, and stability [140]. Additionally, there is a growing focus on adaptive regulatory frameworks that incorporate real-world evidence and post-approval monitoring to address uncertainties associated with novel nanomaterials.

### 5.2. Ethical Considerations

The capacity of NPs to cross the BBB and modulate neural function raises ethical concerns about their potential misuse. For example, theranostic NPs with integrated imaging and drug delivery functionalities could be repurposed for surveillance or cognitive manipulation, threatening individual autonomy and privacy. Furthermore, the long-term effects of NPs on brain development, plasticity, and behavior remain unexplored, raising critical questions about unintended consequences [141].

Transparent communication about the risks and benefits of nanomedicine is crucial to fostering public trust. Interdisciplinary collaborations among ethicists, scientists, and policymakers are needed to establish safeguards against misuse while ensuring equitable access to transformative therapies [142]. The high costs associated with NP development and production raise concerns about accessibility, particularly for underserved populations. Without proactive strategies to reduce costs, advanced nanoparticle-based treatments risk deepening global health inequities [143].

Efforts to democratize access to nanomedicine include leveraging regional manufacturing hubs to reduce logistical costs, adopting open-source platforms for nanoparticle design, and encouraging public–private partnerships to subsidize development. Additionally, simplifying nanoparticle synthesis through scalable, cost-effective techniques—such as bioinspired self-assembly—could significantly lower barriers to entry for emerging markets [144].

### 5.3. Future Perspectives

#### 5.3.1. Advancing Nanoparticle Engineering

Nanoparticle design is transitioning toward multifunctionality, precision, and adaptability. Self-assembling NPs, which dynamically form in response to specific physiological conditions, are emerging as a key innovation. These systems can integrate multiple therapeutic and diagnostic capabilities into a single platform while responding intelligently to environmental cues [145]. For example, NPs that disassemble in response to low pH, releasing their cargo exclusively in acidic tumor or ischemic regions, exemplify the growing sophistication of targeted delivery systems [146,147].

Another frontier is the development of biomimetic NPs that replicate the structure and behavior of natural biological systems. Exosome-mimetic NPs, functionalized with cell-specific surface markers, have demonstrated remarkable success in delivering CRISPR-Cas9 systems for gene editing in neurodegenerative disease models [148]. Hybrid NPs, which combine synthetic materials with bioactive components, are also gaining traction, offering enhanced biocompatibility and tunable therapeutic properties [149].

#### 5.3.2. NPs in Systems Neuroscience

An intriguing direction for future research involves the integration of nanoparticle technologies with systems neuroscience to modulate and repair neural networks. Magnetic NPs, for instance, are being explored for their ability to interface with neural circuits non-invasively, providing both therapeutic and diagnostic functions [150]. Recent studies have demonstrated the potential of magnetic NPs to stimulate neurons selectively, enabling the restoration of disrupted neural pathways in conditions like spinal cord injury or traumatic brain injury [151].

Theranostic NPs equipped with neuroelectronic interfaces are another area of innovation. These systems can bridge damaged neural circuits while providing real-time feedback on neural activity, enabling dynamic adjustment of therapeutic strategies. This convergence of nanotechnology and neuroscience promises to redefine the treatment landscape for complex CNS disorders [152].

#### 5.3.3. Personalized Nanomedicine

The integration of artificial intelligence (AI) and big data analytics into nanoparticle research is accelerating the realization of personalized nanomedicine. AI algorithms can predict optimal nanoparticle designs for individual patients based on their genetic, molecular, and environmental profiles. For example, AI-driven modeling has been used to optimize the surface functionalization of NPs for targeting specific mutations in glioblastoma [153].

Nanoparticle-enhanced liquid biopsies are emerging as a cornerstone of precision medicine. These platforms utilize NPs to isolate and detect circulating biomarkers, such as extracellular vesicles or tumor DNA, from biofluids like blood or cerebrospinal fluid. This real-time monitoring capability enables clinicians to tailor treatments dynamically, improving outcomes and reducing side effects [154].

## 6. Conclusions

### 6.1. Summary of Key Insights

Nanotechnology has emerged as a transformative force in neuroscience, addressing challenges that have long impeded progress in the diagnosis and treatment of CNS disorders. This review underscores the unparalleled versatility of NPs, which overcome the restrictive BBB and leverage precise targeting to deliver therapeutic and diagnostic interventions for conditions such as neurodegenerative diseases, glioblastoma, stroke, and spinal cord injuries. Through innovations in passive and active targeting, stimuli-responsive systems, and bioactive platforms, NPs are redefining the possibilities in CNS medicine.

Theranostic NPs stand out as a pioneering technology, combining imaging and therapeutic capabilities within a single system. These multifunctional platforms enable dynamic tracking of disease progression and therapeutic efficacy, making personalized medicine a reality. Similarly, regenerative applications of NPs in neuroprotection and neural repair are breaking new ground by creating permissive microenvironments for axonal regrowth, reducing oxidative damage, and modulating inflammation to preserve neuronal integrity.

While these advances represent monumental progress, challenges related to safety, scalability, and regulatory approval remain substantial. The complexity of nanoparticle interactions with the CNS, coupled with their unique physicochemical properties, demands new frameworks for toxicity evaluation, manufacturing protocols, and approval pathways. Addressing these challenges will require a concerted effort from researchers, clinicians, and policymakers to ensure that these innovations can transition from laboratory breakthroughs to real-world applications.

### 6.2. Vision for the Future

The future of neuroscience will be shaped by the seamless integration of nanotechnology with emerging fields such as artificial intelligence (AI), systems neuroscience, and regenerative medicine. NPs are no longer just vehicles for drug delivery; they are becoming dynamic tools capable of interacting with their environment, adapting to pathological conditions, and enabling real-time monitoring of disease processes.

Next-generation NPs are likely to incorporate self-assembling properties and biomimetic features, allowing them to navigate complex neural microenvironments autonomously. For instance, hybrid NPs that combine the targeting precision of antibodies with the dynamic adaptability of exosomes could revolutionize the treatment of diseases like Alzheimer’s or glioblastoma, where heterogeneity and disease progression pose significant barriers. Additionally, multifunctional NPs that combine CRISPR-Cas9 gene-editing tools with imaging agents will allow clinicians to intervene at the genetic level while visualizing therapeutic effects in real time.

In the realm of neural repair, magnetic and bioactive NPs are poised to redefine recovery strategies for conditions such as traumatic brain injuries and spinal cord damage. These particles not only deliver therapeutic agents but also guide cellular alignment, promote angiogenesis, and provide mechanical stimulation to encourage neural regrowth. Innovations such as 3D nanostructures and scaffolds, integrated with NPs, offer unprecedented opportunities for creating regenerative environments that support both neural and vascular integration.

The convergence of nanotechnology and neuroelectronics presents another exciting frontier. Neural interface NPs capable of bridging disrupted circuits could enable treatments that dynamically adapt to neural activity, addressing conditions such as epilepsy, stroke, and neuropsychiatric disorders. These systems promise to combine therapeutic intervention with real-time monitoring, advancing our understanding of complex neural networks and their repair.

### 6.3. Further Research and Collaboration

Despite these remarkable advancements, realizing the full potential of nanotechnology in neuroscience requires interdisciplinary collaboration and sustained innovation. Researchers must continue to refine nanoparticle design to address issues of long-term safety, degradation, and immunogenicity. Engineers must develop scalable manufacturing processes that ensure consistency and affordability, while regulatory agencies must adapt their frameworks to accommodate the unique properties of nanomaterials.

Ethical considerations must remain at the forefront of these developments. Ensuring equitable access to nanoparticle-based therapies is critical to avoiding disparities in healthcare. Public engagement will be essential to demystify nanotechnology, addressing concerns about safety and environmental impact while fostering acceptance of these transformative therapies.

Nanotechnology offers a future in which the diagnosis, treatment, and monitoring of CNS disorders are more precise, effective, and personalized than ever before. By integrating advances in biomaterials, molecular engineering, and computational technologies, NPs are poised to redefine the boundaries of neuroscience. Achieving this vision will demand bold innovation, rigorous collaboration, and a commitment to ensuring that these breakthroughs benefit all patients, globally. With sustained effort, nanomedicine promises to transform how we approach the most intractable neurological challenges, opening doors to a new era of therapeutic possibilities.

## Figures and Tables

**Figure 1 ijms-25-13302-f001:**
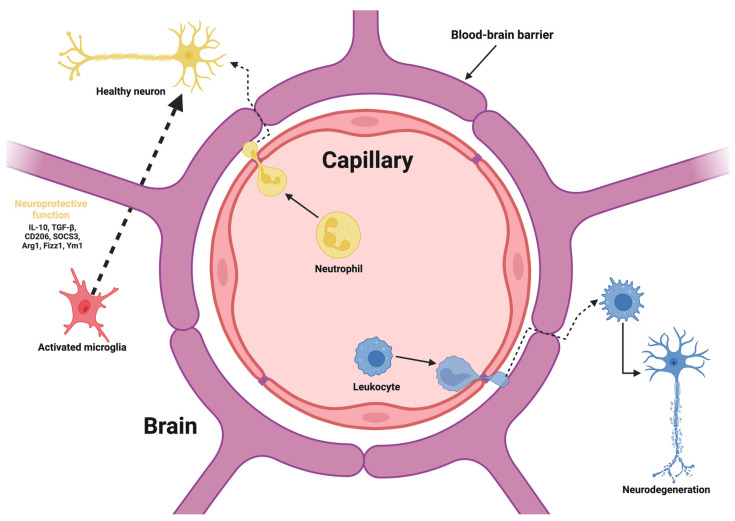
This schematic illustrates the structural and functional dynamics of the BBB and its role in neuroinflammation and neurodegeneration.

**Figure 2 ijms-25-13302-f002:**
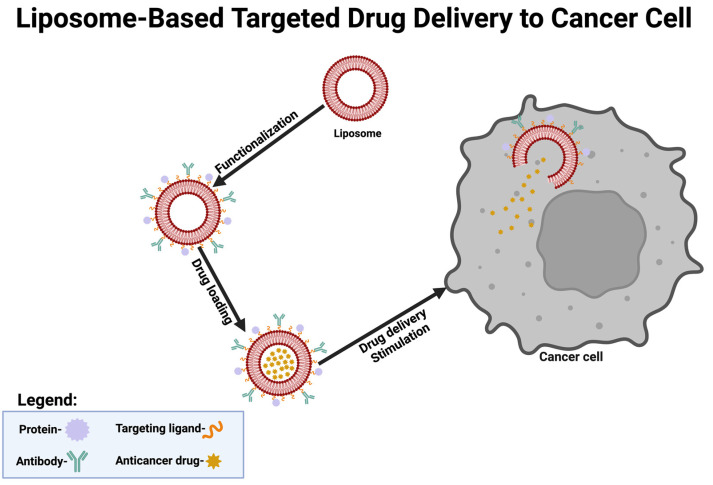
Demonstrates the mechanism of liposome-based targeted drug delivery to cancer cells. Liposomes are functionalized with targeting ligands and antibodies for receptor-specific binding, loaded with anticancer drugs, and delivered to cancer cells.

**Table 1 ijms-25-13302-t001:** Provides an in-depth overview of lipid, polymeric, magnetic, exosome, and hybrid NPs designed to overcome the challenges posed by the BBB. It emphasizes their multifaceted applications and recent advancements in CNS therapies.

Type of Nanoparticle	Mechanism of BBB Penetration	Therapeutic Applications	Diagnostic Applications	Recent Innovations	Advantages	References
Lipid NPs	Passive diffusion; PEGylation extends circulation and reduces immunogenicity.	Delivery of nucleic acids for genetic diseases; chemotherapeutics for glioblastoma.	MRI contrast via lipid-functionalized particles in Alzheimer’s imaging.	mRNA delivery for Huntington’s; functionalized lipid systems for drug bioavailability.	Biocompatibility; adaptable for hydrophilic and hydrophobic drugs.	[17,33,34]
Polymeric NPs	Receptor-mediated transport via ligands (e.g., transferrin and insulin receptors).	Neuroprotective agents for stroke; anti-inflammatory drugs for neurodegeneration.	Real-time inflammation imaging; labeled particles for BBB permeability tracking.	Polymeric scaffolds releasing growth factors for SCI; dual-release drug systems.	Sustained and localized release; customizable for multiple diseases.	[35,36,37]
Magnetic NPs	External magnetic guidance; hyperthermia facilitates localized delivery.	Drug-loaded particles for glioblastoma; neurostimulation in spinal cord injuries.	MRI/CT for glioblastoma; tracking regenerative outcomes in SCI.	High-resolution MRI-enhanced magnetic particles; responsive drug release in ischemia.	Precise spatial control; dual diagnostic and therapeutic functionalities.	[38,39,40]
Exosomes	Endogenous vesicles with intrinsic BBB crossing; modifiable for payloads.	siRNA for Alzheimer’s; protein delivery in Parkinson’s models.	Targeted exosome-based biomarkers for early Alzheimer’s detection.	Exosome engineering for genetic therapies; combined CRISPR/exosome platforms.	Minimal immune response; intrinsic biological compatibility.	[41,42,43]
Hybrid NPs	Combination of passive and active targeting; integrated functional features.	Dual therapies combining imaging and drug delivery for neuro-oncology.	Multimodal imaging for tumor localization and therapeutic evaluation.	Gold–polymer hybrids for photothermal therapy and diagnostics in CNS tumors.	Multifunctional design for complex CNS disorders.	[44,45,46]

**Table 2 ijms-25-13302-t002:** Highlights innovative theranostic nanoparticle systems such as iron oxide, quantum dots, gold, hybrid, and lipid-coated magnetic NPs, providing insights into their dual functionalities in neurology.

Nanoparticle Type	Diagnostic Modality	Therapeutic Modality	Recent Innovations	Advantages	References
Iron Oxide NPs	MRI enhancement for amyloid plaques in Alzheimer’s and tauopathy.	Anti-inflammatory drugs for stroke; enhanced delivery for glioblastoma therapy.	Ultra-small particles for dual diagnostic/therapeutic integration in Alzheimer’s.	High-resolution MRI contrast; superior biocompatibility.	[60,61]
Quantum Dots	Fluorescent imaging for deep-tissue neural damage; dopamine signaling visualization.	Delivery of neuroprotective agents; gene editing with fluorescent tracking.	Near-infrared emission for neurodegenerative disease therapy and tracking.	Exceptional photostability; ability to target small neuronal clusters.	[62,63,64]
Gold NPs	CT and optical imaging of tumor boundaries; tumor growth monitoring.	Photothermal therapy in glioblastoma; chemotherapy in localized CNS tumors.	Gold–lipid hybrids for photothermal and immunotherapy in glioblastoma.	Multimodal imaging and treatment flexibility; precise photothermal control.	[65,66,67]
Hybrid NPs	Dual PET/MRI imaging; real-time biodistribution mapping.	CRISPR-Cas9 delivery for genetic diseases; combination drug release systems.	Multifunctional systems for gene therapy, imaging, and therapeutic release.	Integrative design for personalized treatments and multimodal feedback.	[68,69,70]
Lipid-Coated Magnetic NPs	Ultrasound-responsive imaging of ischemic regions; BBB permeability tracking.	Combined hyperthermia and drug release in ischemia; regeneration tracking in SCI.	Combined photothermal and MRI-targeted therapy in complex CNS conditions.	Efficient delivery across barriers; enhanced tracking and control mechanisms.	[71,72,73]

**Table 3 ijms-25-13302-t003:** Elaborates on cerium oxide, chitosan, mesoporous silica, exosome-mimetic, and peptide-functionalized magnetic NPs, illustrating their critical roles in CNS repair and recovery.

Nanoparticle Type	Neuroprotective Mechanism	Regenerative Applications	Recent Innovations	References
Cerium Oxide NPs	Persistent ROS scavenging; long-term oxidative stress mitigation.	Stroke therapy with neural recovery; sustained antioxidant control.	Improved catalytic activity for longer therapeutic effects.	[89,105,106]
Chitosan NPs	Anti-inflammatory modulation of microglia; enhanced neuronal survival.	SCI therapy with growth factor delivery and reduced gliosis.	Dual-action systems combining anti-inflammatory and regenerative agents.	[107,108,109]
Mesoporous Silica NPs	Neural stem cell differentiation promotion; angiogenesis stimulation.	Axonal regrowth and vascularization support in chronic CNS injuries.	Functionalization with ligands for targeting and controlled differentiation.	[110,111,112]
Exosome-Mimetic NPs	Delivery of miR-124 to reduce glial scarring and promote neurogenesis.	Ischemic recovery with neurogenesis stimulation and vascular integration.	Exosome-like NPs for precision delivery and genetic repair.	[113,114,115]
Peptide-Functionalized Magnetic NPs	Axonal alignment and stimulation; protective factor release.	Neural circuit realignment; combined regenerative therapy and imaging.	Advanced targeting with peptides for guided neural regeneration.	[116,117,118]

## Data Availability

Not applicable.

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
