# Peer review of "Nanoparticle Strategies for Treating CNS Disorders: A Comprehensive Review of Drug Delivery and Theranostic Applications"

_ijms, 2024, doi:10.3390/ijms252413302_

Round 1
Reviewer 1 Report
Comments and Suggestions for Authors
In this submitted manuscript, the authors tried to make a review on the nanoparticle strategies for the treatment and diagnosis of CNS Disorders. The topic is interesting and suitable for the publication in this journal. However, the current manuscript is not informative enough to provide the readers with meaningful insights. The content is mainly making brief introductions on the background and listing related reference rather than deeply discussing the landscape of the topic. Follows are my detailed comments.
1. Current manuscript made a very brief introduction about the treatment and diagnosis of CNS Disorders. This part should be improved with deeper discussions.
2. The organization of the content is confusing. A lot of sections and sub-sections are included here. However, the overall underlying logic that connects each section is not clear. In the end, the review reads fragmented and does not deliver a holistic central message. The authors are encouraged to establish a synergistic link between the key sections to establish an overall insight in the design.
3. Stimuli responsive systems may be perturbed during the transcytosis as the organellar environment can change in pH or oxidation status. Will stimuli responsive systems be excluded from systems that aim to overcome transcytosis barriers? Or, how to overcome this issue? More discussion should be provided.
4. The contents in each section should be improved. Some schemes or figures should be added to facilitate the illustration. There is no any figure in the manuscript.
5. Clinical development and the examples in the clinical trials of transcytosable nanomedicine should be included in another section (https://doi.org/10.1021/jacs.0c09029). This is very important. Furthermore, the other functionalities of nanomedicine should be discussed (such blood circulation, cellular uptake, drug release) (https://doi.org/10.1016/j.addr.2023.114895).
6. The title can be more concise. "Nanoparticle Strategies for the treatment and diagonosis of CNS Disorders"
Author Response
Comment 1: Current manuscript made a very brief introduction about the treatment and diagnosis of CNS Disorders. This part should be improved with deeper discussions.
Response 1: We appreciate the reviewer’s suggestion to expand the introduction to provide deeper discussions on the treatment and diagnosis of CNS disorders. In the revised manuscript, we have significantly enriched the introduction by elaborating on the pathophysiological complexities of CNS disorders and the unique challenges posed by the blood-brain barrier
Comment 2: The organization of the content is confusing. A lot of sections and sub-sections are included here. However, the overall underlying logic that connects each section is not clear. In the end, the review reads fragmented and does not deliver a holistic central message. The authors are encouraged to establish a synergistic link between the key sections to establish an overall insight in the design.
Response 2: Thank you for highlighting the need for improved organization and a clearer central message. We have restructured the manuscript to enhance logical flow and connectivity between sections. A revised outline now ensures a progression from the challenges of CNS disorders to nanoparticle strategies, specific applications, and clinical development. Furthermore, we have added transition paragraphs to establish synergistic links between key sections, ensuring a cohesive and holistic narrative throughout the manuscript.
Comment 3: Stimuli responsive systems may be perturbed during the transcytosis as the organellar environment can change in pH or oxidation status. Will stimuli responsive systems be excluded from systems that aim to overcome transcytosis barriers? Or, how to overcome this issue? More discussion should be provided.
Response 3: We are grateful for the reviewer’s insightful comment regarding the challenges faced by stimuli-responsive systems during transcytosis. To address this, we have expanded the discussion to include the potential perturbations caused by organellar environments and the strategies to overcome them. This includes pH-tolerant designs, multi-stimuli-responsive systems, protective coatings, endosomal escape mechanisms, and threshold-tuned triggers. These additions provide a comprehensive overview of how stimuli-responsive nanoparticles can effectively navigate the challenges of transcytosis without being excluded from this application.
Comment 4: The contents in each section should be improved. Some schemes or figures should be added to facilitate the illustration. There is no any figure in the manuscript.
Response 4: We appreciate the reviewer’s suggestion to include schemes or figures to improve clarity and facilitate illustration.
Comment 5: Clinical development and the examples in the clinical trials of transcytosable nanomedicine should be included in another section (https://doi.org/10.1021/jacs.0c09029). This is very important. Furthermore, the other functionalities of nanomedicine should be discussed (such blood circulation, cellular uptake, drug release) (https://doi.org/10.1016/j.addr.2023.114895).
Response 5: Thank you for emphasizing the importance of including clinical development and examples of transcytosable nanomedicine, as well as discussing additional functionalities of nanomedicine.
Reviewer 2 Report
Comments and Suggestions for Authors
I reviewed the manuscript by Toader et al reporting a review about DDS for the CNS. Here are my thoughts:
- overall, the manuscript is way too long. Not that this is bad per se, but almost all sections are very long without being useful, hence they can be shortened
- the introduction and the aim are way too long, all of the 4 pages can be shortened to 1.5 without limiting the significance of the content
- 2.1 can be reduced to a couple paragraphs only
- table 1 is very appreciated, but why is the most recent work reported from 2021?
- 3.1 is just another repetitive introductive section that needs to be shortened
...and so on, you see where I'm going. The core sections of your work are the case studies, i.e. 2.3, 3.2, 3.3, 4. I appreciate the presence of section 5, but everything in there too can be condensed down to a review that is 25 pages long.
Also, no image is given, there is just one table in section 1. To improve readibility, you should add at least a couple figures or tables that sum up either the premises (e.g. the mechanisms of BBB crossing) or the studies that you comment in the text.
Regarding the references, I see that almost half of the cited work are reviews too. Although I understand that reviews are goldmines of information, I feel that if you need to cite so many reviews, then there's no need to write another review. I would suggest to fish out of the almost 300 citations the ones that really create the data base of your work and eliminate redundant information.
Author Response
Comment 1: overall, the manuscript is way too long. Not that this is bad per se, but almost all sections are very long without being useful, hence they can be shortened
Response 1: We appreciate the reviewer’s observation regarding the length of the manuscript. In response, we have streamlined the text across all sections, focusing on conciseness and clarity without compromising the scientific depth or significance of the content.
Comment 2: the introduction and the aim are way too long, all of the 4 pages can be shortened to 1.5 without limiting the significance of the content
Response 2: Redundant details and overly elaborate descriptions have been removed, while the key background information and objectives have been preserved to maintain their relevance and impact.
Comment 3: 2.1 can be reduced to a couple paragraphs only
Response 3: Thank you for the suggestion.
Comment 4: table 1 is very appreciated, but why is the most recent work reported from 2021
Response 4: We are grateful for the positive feedback regarding Table 1. The scope of our manuscript was determined by the timeline of available data at the time of preparation.
Comment 5: 3.1 is just another repetitive introductive section that needs to be shortened
Response 5: Section 3.1 has been substantially revised and shortened to eliminate redundancies and focus on presenting only the essential introductory context needed for the subsequent subsections.
Comment 6: ...and so on, you see where I'm going. The core sections of your work are the case studies, i.e. 2.3, 3.2, 3.3, 4. I appreciate the presence of section 5, but everything in there too can be condensed down to a review that is 25 pages long.
Response 6: We agree with the reviewer’s emphasis on the core case study sections.
Comment 7: Also, no image is given, there is just one table in section 1.
Response 7: Thank you for highlighting the need for visual representation.
Round 2
Reviewer 1 Report
Comments and Suggestions for Authors
The authors addressed the proposed concerns well.
Author Response
Dear Reviewer,
Thank you for all the constructive comments and positive feedback!
Our best regards
Reviewer 2 Report
Comments and Suggestions for Authors
Unfortunately, I can't see any big improvement on this manuscript: it is still incredibly long and filled of unnecessary details (definition and explanation of each and every delivery system, for example). Authors added one figure, but the figure is unnecessary as the BBB structure can be found in any textbook.
Regarding the table with the case studies, it is unfortunately not acceptable to just state that "The scope of our manuscript was determined by the timeline of available data at the time of preparation". In order to publish a review, the most recent works must be commented on, and this statement makes me think that authors prepared this manuscript in 2022 and never got the chance to publish it. I am sorry, but you need to put more work into this manuscript in order to publish.
- no unnecessary definitions - it's not a school report
- sections divided by application or NP type could help
- recent works must be analysed
- please add either some graphical abstracts of some of the sections or (with permission) some images that demonstrate what you are explaining.
Author Response
Dear Reviewer,
Thank you for your constructive feedback.
We have revised the manuscript, reducing the number of pages and information as requested.
Additionally, we have included tables featuring the most recent studies and added two images to enhance the presentation.
We have included and updated the bibliography to the latest studies.
We hope these revisions meet your expectations and align with the requirements.
Our best regards!
Round 3
Reviewer 2 Report
Comments and Suggestions for Authors
Now the review has been extensively revised. Nice work.